# Polarity-Aware Semantic Retrieval with Fine-Tuned Sentence Embeddings

## Abstract

This paper investigates the effectiveness of retrieving sentences with multiple objectives – polarity and similarity – by fine-tuning sentence-transformer models on augmented supervised data. We establish two opposing metrics, namely Polarity Score and Semantic Similarity Score, for evaluation purposes. These are used in a test suite with various lightweight sentence-transformer models, hyperparameters and loss functions. Experiments are conducted on two binary classification problems from different domains: the SST-2 dataset for sentiment analysis and the detection of sarcastic news headlines. Addressing the catastrophic forgetting problem, our results show that the configuration of loss functions drastically alters a model's capability to retain similarity while simultaneously differentiating on classes from supervised data. These findings indicate that we can 1) improve upon generalized sentence embeddings for information retrieval and 2) increase interpretability of sentence embeddings by studying their adaptability to different domains.

## 1 Introduction

In the rapidly evolving field of Natural Language Processing, the tasks of text classification and semantic textual similarity (STS) are well established and have countless use cases. While rule-based, statistical and deep learning models for both tasks have been successful throughout the years (Tai et al., 2015; Minaee et al., 2021; Li et al., 2022), newer contextual word representations and transformer models have now become the de-facto standard (Joulin et al., 2017; Howard & Ruder, 2018; Devlin et al., 2019; Yang et al., 2019; Raffel et al., 2020).

Sentence embeddings have also shown great promise for STS (Reimers & Gurevych, 2019), often trained by contrastive learning (Chuang et al., 2022; Gao et al., 2022). Research suggests these procedures are effective with much less data than previously needed for end-to-end models, as shown with few-shot training examples in SetFit (Tunstall et al., 2022). By incorporating classification into the data sources for sentence-transformers and adjusting the training configurations, we study the capability of restructuring the embedding space throughout fine-tuning to capture both sentences of the same polarity and of high semantic similarity. This scheme also allows for standard classification by considering the labels of retrieved similar sentences in the training data.

For evaluation, we establish two metrics: Polarity Score, which measures the classification performance, and Semantic Similarity Score, which quantifies the semantic closeness of texts compared to a reference model. These metrics allow us to closely interpret the behavior of the resulting semantic space in different domains, addressing the problem of catastrophic forgetting during fine-tuning (Goodfellow et al., 2015; Opitz & Frank, 2022).

Experiments are conducted on two datasets: 1) SST-2, Stanford Sentiment Treebank (Socher et al., 2013), a binary sentiment dataset on full sentences, and 2) A dataset with sarcastic news headlines (Misra & Arora, 2023). The remainder of this paper is structured as follows: Section 2 discusses related work. Section 3 introduces the datasets, metrics, models and training details. Section 4 presents experimental results and Section 5 discussions. Finally, conclusions and future work are described in Section 6.

## 2 Related Work

Related research is largely based on developments within word and sentence embeddings. Commonly used embedding techniques include word2vec (Mikolov et al., 2013), GloVe (Pennington et al., 2014), and ELMo (Peters et al., 2018). In the realm of sentence embeddings, early methods involved concatenation and aggregation of word embeddings to produce a sentence representation (Le & Mikolov, 2014; Joulin et al., 2017). However, more recent research has focused on developing specialized models to encode sentence representations, as exemplified by systems like InferSent (Conneau et al., 2017), universal sentence encoder (Yang et al., 2020), sentence-transformers (SBERT) (Reimers & Gurevych, 2019) and SimCSE (Gao et al., 2022). SBERT is trained using a pre-trained BERT model to learn the representations of a given sentence. While techniques and setups vary, an example of a training procedure is by providing triplets forming *(anchor sentence, positive, negative)*, where the model attempts to maximize the distance between the anchor and the *negative* (dissimilar sentence), while minimizing the distance between the anchor and the *positive* (similar) sentence. This methodology provided efficient models for STS (Agirre et al., 2013; Reimers & Gurevych, 2019; Gao et al., 2022; Tunstall et al., 2022; Wang et al., 2022; Li et al., 2023). Several datasets and benchmarks have been published for STS since the SemEval shared task (Agirre et al., 2013), including the *STS Benchmark* (Cer et al., 2017), SICK (Marelli et al., 2014), and BIOSSES (Soğancıoğlu et al., 2017), all of which are now found in the Massive Text Embedding Benchmark (MTEB) (Muennighoff et al., 2022). Transformer models have excelled at the task, as is shown in the tables on HuggingFace's leaderboard for the evaluation.[1] Currently, the *General Text Embeddings* model (Li et al., 2023) receives the highest scores. The work by Opitz & Frank (2022) is highly related to interpretability for multiple objectives, where the authors create a set of sub-embeddings for features such as negation and semantic roles, adressing the problem catastrophic forgetting (Goodfellow et al., 2015). This problem has been further studied in detail by Chen et al. (2020), adjusting the mechanisms behind the Adam optimizer (Kingma & Ba, 2017), and Luo et al. (2023), a study describing the forgetting effect during fine-tuning of large language models on various key features like domain knowledge and reasoning. In this work, however, the focus is shifted towards understanding the embedding space for specific domains by augmenting the data sources directly and adjusting the parameters behind the loss functions.

## 3 Methods and Data

This section includes information on datasets, evaluation metrics, baseline models, loss functions, data generation, and the fine-tuning pipeline. We use two sources for classification evaluation. The modeling scheme is generalized to any data source for binary classification.

**SST-2** The Stanford Sentiment Treebank (Socher et al., 2013) is commonly used for binary classification tasks and is implemented in the GLUE benchmark (Wang et al., 2019). It consists of a train/test/validation split with 67,349/1821/872 samples respectively. However, the labels for the test split are hidden and can only be evaluated by submissions to GLUE.[2] As our system is not aimed at the broad range of tasks present in GLUE, we evaluate using the available validation split, for which our system achieves an accuracy of 93.23. A high classification score is not the purpose of this work and is merely an indicator of how retrieved similar sentences can be used to infer the label of an unseen sentence.

**Sarcastic Headlines** The "News Headlines Dataset for Sarcasm Detection" (Misra & Arora, 2023) contains 28,619 news headlines from *HuffPost* (non-sarcastic) and *The Onion* (sarcastic). Misra & Arora claims this to guarantee high quality labels. Furthermore, headlines are primarily self-contained and do not rely on additional context, thus well suited for evaluating both similarity and polarity. Retrieving similar sarcastic sentences to produce labels for the test set gives an accuracy of 92.27, outperforming the models presented by Amin et al. (2023).

---

[1]`https://huggingface.co/spaces/mteb/leaderboard`
[2]`https://gluebenchmark.com/leaderboard`

## 3.1 EVALUATION

For a sentence $s$, we retrieve the $k$ nearest neighbours with a model $M$, denoted $s_1^M, \ldots, s_k^M$. These are evaluated on the criteria of polarity and semantic similarity.

### 3.1.1 POLARITY SCORE

To measure whether a model favors texts of the same polarity as the input in its predictions, we compute a weighted average polarity score over the $k$ predictions depending on the polarity of $s$. Formally, for a sentence $s$, this can be expressed as:

$$\mathcal{P}_M(s) := \sum_{i=1}^{k} w_i \cdot \mathrm{pol}\left(s_i^M\right) \ \text{ where } \ \mathrm{pol}\left(s_i^M\right) := \begin{cases} 1 & \text{if } s \text{ and } s_i^M \text{ have the same polarity,} \\ 0 & \text{otherwise.} \end{cases}$$

(1)

The weights $w_i$ can be chosen to reflect the importance of ranked suggestions. Instead of averaging them, we choose a linear discounting model where the $i$-th suggestion is scaled by a factor of $k + 1 - i$. By normalization, we get weights $w_i := \frac{2 \cdot (k+1-i)}{k \cdot (k+1)}$.

If the predictions are mostly of the same polarity as the input, this is reflected in a value close to one. In any case, we would expect fine-tuned models to be better at predicting sentences of the same polarity than the pre-trained baseline, or *reference*, model.

### 3.1.2 SEMANTIC SIMILARITY SCORE

In assessing the quality of predicted sentences, simply aligning their polarity with the input is insufficient. We necessitate a metric to gauge the semantic similarity: the weighted average cosine similarity between the predictions from a model $M$ and their corresponding embeddings under a baseline reference model $R$, pre-trained for semantic similarity. As the fine-tuned model will increase its internal representation of similarity within its embeddings throughout training, it is necessary to compare similarity with a reference model. The cosine similarity is defined as $\mathrm{cos\_sim}(s_1, s_2) := \frac{x_1 \cdot x_2}{||x_1|| \cdot ||x_2||}$, where $x_i$ is the vector for sentence $s_i$. For a model $M$, we compute the Semantic Similarity Score $\mathcal{S}_M$ for a sentence $s$:

$$\mathcal{S}_M(s) := \sum_{i=1}^{k} w_i \cdot \mathrm{cos\_sim}_R\left(s, s_i^M\right)$$

(2)

The weights $w_i$ are reused from the Polarity metric, as defined in Section 3.1.1. If the predicted sentences from model $M$ remain semantically similar to the input sentence, we should observe that $\mathcal{S}_M(s)$ is equal or slightly lower than the reference $\mathcal{S}_R(s)$.

## 3.2 BASELINE MODELS

The models in Table 1 are selected based on varying complexity, but more importantly, performance versus size and inference time. Data is sourced from the MTEB leaderboard (Muennighoff et al., 2022). We select the commonly used sentence-transformer model, *all-MiniLM-L6-v2* (Reimers & Gurevych, 2019) – referred to as *MiniLM-6*, along with the better performing models *GTE-base/small* (Li et al., 2023) and the *E5-small-v2* (Wang et al., 2022). We use the entire test sets for target embeddings, and select a lookup sample of five times the size of the test set as source embeddings – keeping an even ratio between datasets. Inspecting the importance of $k$ for each model shows a near static relationship between the models (see Figure 1), where performance drops slightly for higher values of $k$, as can be expected when forcing the model to retrieve more sentences. We select $k = 16$ for further experiments, as a reasonable number for retrieval and inspection, as well as to reduce the number of computations. Although performance is generally high for all values, the *E5-small* model achieves the highest scores. Conversely, the *minilm-6* performs the worst, with especially low scores for $\mathcal{S}$. All models are used for continued evaluations.

Table 1: Sentence-transformer baseline model selection and performance ($k = 16$) for polarity and semantic similarity on SST-2 and sarcastic headlines. Standard deviation subscripted.

| Model | Size MB | Embedding dimension | STSBenchmark reported avg | SST-2 $\mathcal{P}$ | SST-2 $\mathcal{S}$ | Sarcasm $\mathcal{P}$ | Sarcasm $\mathcal{S}$ |
|---|---|---|---|---|---|---|---|
| E5-small-v2 | 130 | 768 | **85.95** | $\mathbf{81.5_{23.7}}$ | $\mathbf{85.5_{1.7}}$ | $\mathbf{71.4_{21.2}}$ | $\mathbf{83.4_{1.5}}$ |
| GTE-base | 220 | 768 | 85.73 | $80.4_{22.6}$ | $83.7_{1.4}$ | $67.4_{20.7}$ | $81.4_{1.6}$ |
| GTE-small | 70 | 384 | 85.57 | $77.8_{22.2}$ | $84.8_{1.4}$ | $66.8_{20.6}$ | $82.5_{1.6}$ |
| MiniLM-6 | 90 | 384 | 82.03 | $63.0_{21.9}$ | $46.6_{7.4}$ | $63.8_{20.2}$ | $42.3_{5.6}$ |

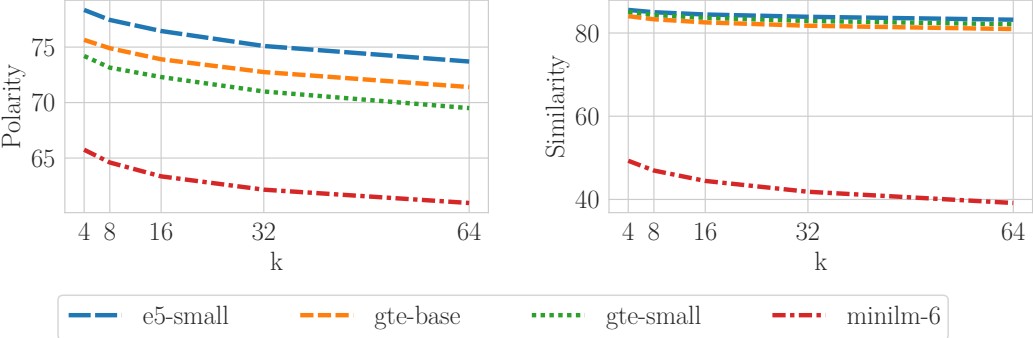

Figure 1: Baseline models with average performance across both datasets when retrieving the $k$ nearest matches.

### 3.3 Loss Functions for Sentence Embeddings

To assess the quality of sentence embeddings, models are trained with different loss functions depending on the desired properties for downstream tasks. The Sentence-Transformers library provides a wide range of predefined loss functions.[3] However, not all losses provide the desired flexibility for supporting our constraints of multiple objectives. There are four batching triplet losses, all of which generate every valid combination of triplets, typically creating a far too large dataset. We describe triplets and their generation constraints in Section 3.4. Further, the *DenoisingAutoEncoderLoss* adds noise and reconstructs the original sentences. This process is suitable for unsupervised training, but not for our application of encoding polarity within the embeddings. The same limitation holds for *MSELoss*, which uses the MSE loss between a target and source, with no relation to it being positive or negative. *MegaBatchMarginLoss* finds the least similar pair between an anchor and a sentence of the same polarity. As our similarity scores are not gold labels, we find this loss incompatible. *MarginMSELoss* requires a gold similarity score between a query and a positive/negative value, which we do not have. *CosineSimilarityLoss* considers the similarity between pairs of sentences. This is the very basis for the augmentation of datasets to begin with, as we have ensured a threshold of similarity between the sentences of equal polarity. However, this loss is the default for SetFit (Tunstall et al., 2022), which we use in our comparisons. The *SoftMaxLoss* was in Reimers & Gurevych (2019) used to train models on NLI data (Williams et al., 2018), adding a softmax classifier on the output, compatible with multiple classes. However, it does not provide a clear distinction of *similarity*. After filtering, we employ a set of four loss functions: TripletLoss (Schroff et al., 2015), MultipleNegatives-RankingLoss (Henderson et al., 2017), OnlineContrastiveLoss and ContrastiveLoss (Hadsell et al., 2006). These have varying data inputs related to how the model assesses the similarity between input sentences. All models support a similarity function, for which we use the cosine similarity. They are described in more detail below.

---

[3]We encourage the interested reader to study the loss functions at `https://www.sbert.net/docs/package_reference/losses.html`

**TripletLoss** consists of triplets of sentences $(A, P, N)$ where $A$ is the "anchor", $P$ is similar to the anchor, and $N$ is dissimilar. In context of binary classification, P is attributed to the label 1, and N label 0. The loss is then expressed as: $\max(|\text{emb}(A) - \text{emb}(P)| - |\text{emb}(A) - \text{emb}(N)| + \lambda, 0)$, where $\lambda$ is the margin, specifying the minimum separation between $A$ and $N$.

**MultipleNegativesRankingLoss** consists of sentence pairs, assuming $(a_i, p_i)$ pairs as positive and $(a_i, p_j)$ pairs for $i \neq j$ as negatives. It calculates the loss by minimizing the negative log-likelihood for softmax-normalized scores, encouraging positive pairs to have higher similarity scores than negative pairs.

**(Online)ContrastiveLoss** consists of $\{0, 1\}$-labelled tuples $(\text{Anchor}, \text{Sentence})$ where the label indicates whether $|\text{emb}(A) - \text{emb}(S)|$ is to be maximized, indicating dissimilarity (0) or minimized, indicating similarity (1). In the online variant, the loss is only calculated for strictly positive or negative pairs, reported to generally perform better (Tunstall et al., 2022). The margin parameter $\lambda$ controls how far dissimilar pairs need to be separated.

For each compatible loss function, we select various margin values (Table 2) in order to study the models' behavior.

Table 2: Loss functions with margin selections.

| Loss function | $\lambda$ **margin** | $\lambda$ **default** |
|---|---|---|
| Triplet Loss | $\{0.01, 0.1, 1.0, 5.0, 7.5, 10\}$ | 5.0 |
| Multiple Negatives Ranking Loss | $-$ | $-$ |
| Contrastive Loss | $\{0.1, 0.25, 0.5, 0.75, 1.0\}$ | 0.5 |
| Online Contrastive Loss | $\{0.1, 0.25, 0.5, 0.75, 1.0\}$ | 0.5 |

### 3.4 DATA GENERATION

Different loss functions require different data inputs. To speed up data sampling when training, we precompute datasets corresponding to each input type: 1) Triplet, 2) Contrastive, and 3) MultipleNegatives, referred to as *example generation*. Original data is encoded using a sentence-transformer model, from which an index is built. For each (sentence, label) pair in the data, we compute the $k$ nearest neighbors of each polarity, requiring a minimum semantic similarity threshold of $\geq 0.5$, resulting in semantically similar pairs for each label. These pairs are then combined according to the selection of loss functions, e.g., with a TripletLoss requiring (anchor, similar, dissimilar). As this process creates a mapping between every source sentence to the $k$ similar sentences, we control the data generation size by introducing a *dropout*, tuned to generate roughly 250,000 examples for each loss function, the highest number we can reach when normalizing the sample count across all configurations.[4] See Figure 2 for an illustration of the example generation process. Data examples for each loss type are listed in Table 3.

### 3.5 FINE-TUNING

The generated examples, described in Section 3.4, is the input to each model configuration, forming the basis for fine-tuned models. From the selected dataset, we fetch the generated examples corresponding to the loss function, from which $N$ are resampled each training step. The dataset, in its original form, is passed to the reference model as well as the fine-tuned model after each training step, from which an index is computed to retrieve the $k$ closest matches in the training samples for each sample in the test split, used to compute the scores for polarity and semantic similarity.

---

[4]Data generation for MultipleNegativesRankingLoss with 0 dropout for the smallest dataset (sarcastic headlines, 22,000 samples) produces 243,793 examples.

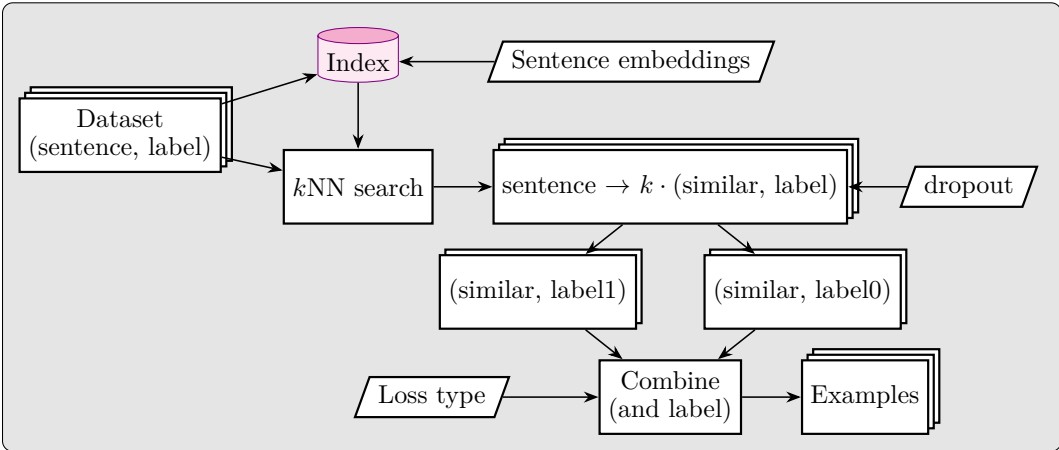

Figure 2: The example generation process.

Table 3: Data samples from SST-2 for the different loss function categories.

| Loss type | Data sample | Data type |
|---|---|---|
| Triplet | **Anchor:** Totally unexpected directions
**Similar+Same polarity:** Dramatically moving
**Similar+Opposite polarity:** Utterly misplaced | Triple |
| Multiple Negatives | **Anchor:** Bring new energy
**Similar+Same polarity:** Juiced with enough energy and excitement | Tuple |
| Contrastive | **Anchor:** Is a movie that deserves recommendation
**Similar:** Effort to watch this movie
**Label:** 0 (increase distance → make less similar)

**Anchor:** Of the jokes, most at women's expense
**Similar:** Dumb gags, anatomical humor
**Label:** 1 (reduce distance → make more similar) | Tuple + Label |

## 4 EXPERIMENTS AND RESULTS

The results are based around fine-tuning and continuous evaluation of the baseline models in different setups for loss functions and corresponding parameters. From available literature, fine-tuning transformers between 1 to 3 epochs seems sufficient in many cases (Gao et al., 2022). Beyond this, we observe smaller improvements – but no signs of overfitting. To decide on a suitable number of training samples (in the range $[50, 100000]$) for further experiments, we study the differences between models after 5 epochs. Despite the reported effectiveness of few-shot learning for sentence-transformers (Tunstall et al., 2022), we observe improvements in polarity when increasing the sample size far beyond the scope of few-shot learning. Table 4 illustrates this behavior, aggregated across all models and loss configurations. While the polarity score $\mathcal{P}$ increases, the semantic similarity score $\mathcal{S}$ takes a slight hit throughout training. The latter is to be expected because we fine-tune the embedding only based on polarity labels. However, the reduction of $\mathcal{S}$ is far lower than the increase in $\mathcal{P}$. Observe the growing distance between the *min* and *max* scores for $\mathcal{S}$. This distance indicates that certain model and loss configurations perform vastly better (or worse) for our joint task, and is the basis for our hypothesis that we can balance both, despite the apparent trade-off. This is further supported by the relatively small changes to the standard deviation. Figure 3 shows an increasing number of outliers as the embedding space is shifted towards polarity, which we aim to minimize with training configurations. To investigate possible configurations, while accounting for computational efficiency, we continue by setting the sample size $N = 50{,}000$ and perform detailed experiments on the aforementioned loss functions with their $\lambda$ margins on both datasets. Details on training and configurations are found in Appendix A.

Table 4: Aggregated scores across all configurations for different sample sizes after 5 epochs on the validation split of the SST-2 dataset.

| Samples | Polarity Score | | | | Semantic Similarity Score | | | |
|---|---|---|---|---|---|---|---|---|
| | **Mean** | $\sigma$ | **Min** | **Max** | **Mean** | $\sigma$ | **Min** | **Max** |
| 50 | 75.7 | 7.5 | 63.0 | 81.5 | **75.1** | **16.6** | **46.6** | **85.5** |
| 500 | 75.7 | 7.5 | 63.0 | 81.5 | **75.1** | **16.6** | **46.6** | **85.5** |
| 2000 | 75.7 | 7.5 | 62.9 | 81.7 | **75.1** | **16.6** | **46.6** | **85.5** |
| 5000 | 76.3 | 7.7 | 63.1 | 83.1 | **75.1** | **16.6** | 46.5 | **85.5** |
| 10000 | 78.0 | 8.3 | 63.2 | 87.3 | 74.9 | 16.8 | 45.7 | 85.4 |
| 20000 | 81.5 | 8.7 | 61.8 | 89.2 | 73.0 | 18.3 | 36.4 | 84.9 |
| 50000 | 86.2 | 6.4 | 68.0 | 92.5 | 70.2 | 21.3 | 29.6 | 84.7 |
| 100000 | **88.9** | **4.0** | **72.2** | **93.4** | 69.3 | 22.3 | 29.0 | 84.6 |

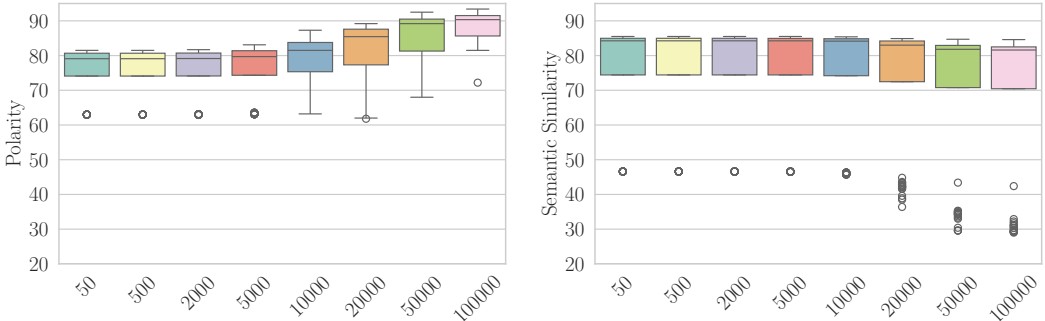

Figure 3: Box plot of polarity- and semantic similarity scores for each sample size on the SST-2 dataset.

Tables 5 and 6 show the polarity and semantic similarity scores obtained after the continued training with $N = 50,000$ samples. The tables are organized to showcase the impact of the different loss functions and their $\lambda$ margins. Best scores are shown in boldface, with the reference model, the setfit baseline, and the best performing model(s) highlighted. Note that for semantic similarity, we boldface the *top two* highest scores, as the MultipleNegative ranking loss – although seemingly performing strongly on the task – does so due to minimal adaptation to the new training samples, with similar performance to the respective baseline models. This can be confirmed by inspecting its polarity scores.

## 5 DISCUSSION

Most model configurations adjusted the embeddings towards correct polarity upon fine-tuning. However, the *minilm-6* falls short of its semantic similarity capabilities, while the remaining models seem to learn both tasks, with only slight differences between configurations. The TripletLoss stands out as the best-performing loss function, especially for smaller margins, with $\lambda \in \{0.01, 0.10\}$, strongly outperforming the default value of 5.0. The earlier referenced statement on OnlineContrastiveLoss generally performing better than ContrastiveLoss holds for most experiments.[5] For the ContrastiveLoss configurations, the default $\lambda$ value of 0.5 seems well suited for the tasks, with minimal changes for different margins. MultipleNegativesRankingLoss is an outlier in both results. This is likely attributed to poor example generation for this particular loss function. MultipleNegatives treats sentences from distinct sentence pairs as dissimilar. In our case, we generate multiple similar pairs with the same first sentence, resulting in contradictory examples. This problem does not arise for any of the other loss functions. The key takeaway is that the implicit relations between distinct training examples severely restrict our flexibility in example generation. Hence,

---

[5]https://www.sbert.net/docs/package_reference/losses.html#onlinecontrastiveloss

Table 5: Polarity scores for all loss configurations after 5 epochs with $N = 50,000$ samples, retrieving $k = 16$ sentences.

| Loss | $\lambda$ | e5-small | | gte-base | | gte-small | | minilm-6 | |
|---|---|---|---|---|---|---|---|---|---|
| | | sarcastic | sst2 | sarcastic | sst2 | sarcastic | sst2 | sarcastic | sst2 |
| Reference | - | $71.4_{21.2}$ | $81.5_{23.7}$ | $67.4_{20.7}$ | $80.4_{22.6}$ | $66.8_{20.6}$ | $77.8_{22.2}$ | $63.7_{20.2}$ | $63.0_{21.9}$ |
| Cosine (SetFit) | - | $85.2_{25.4}$ | $86.2_{24.2}$ | $82.1_{26.8}$ | $85.6_{25.5}$ | $82.8_{25.4}$ | $84.2_{25.9}$ | $79.5_{27.0}$ | $77.9_{29.0}$ |
| Contrastive | 0.10 | $88.8_{24.3}$ | $89.5_{23.2}$ | $86.9_{25.6}$ | $89.2_{24.1}$ | $81.9_{27.0}$ | $88.0_{25.6}$ | $75.9_{25.6}$ | $68.0_{24.9}$ |
| Contrastive | 0.25 | $89.3_{25.1}$ | $90.7_{23.2}$ | $88.2_{26.1}$ | $90.0_{25.0}$ | $84.3_{26.9}$ | $88.8_{26.1}$ | $76.8_{26.4}$ | $72.4_{26.9}$ |
| Contrastive | 0.50 | $89.8_{25.6}$ | $91.2_{23.8}$ | $88.8_{26.5}$ | $90.3_{25.3}$ | $86.8_{27.5}$ | $89.1_{27.1}$ | $77.8_{27.2}$ | $75.1_{27.8}$ |
| Contrastive | 0.75 | $89.9_{25.1}$ | $91.6_{23.6}$ | $88.9_{26.6}$ | $90.6_{25.1}$ | $87.7_{27.3}$ | $89.5_{26.9}$ | $79.0_{27.7}$ | $77.3_{28.6}$ |
| Contrastive | 1.00 | $89.8_{25.5}$ | $91.2_{24.3}$ | $88.7_{26.7}$ | $90.7_{25.1}$ | $87.8_{27.0}$ | $89.6_{26.8}$ | $80.3_{28.1}$ | $78.4_{28.9}$ |
| MultipleNeg | - | $73.6_{22.2}$ | $80.8_{22.4}$ | $73.1_{22.4}$ | $81.8_{23.5}$ | $72.0_{22.6}$ | $80.6_{23.4}$ | $69.0_{22.0}$ | $69.4_{23.1}$ |
| OnlineContr | 0.10 | $89.6_{24.7}$ | $90.4_{23.7}$ | $87.4_{25.8}$ | $89.5_{24.2}$ | $82.6_{27.0}$ | $88.2_{25.8}$ | $78.9_{26.0}$ | $70.8_{26.5}$ |
| OnlineContr | 0.25 | $90.0_{25.2}$ | $91.5_{23.8}$ | $88.2_{26.4}$ | $90.2_{25.4}$ | $84.4_{27.3}$ | $88.9_{26.7}$ | $78.9_{26.4}$ | $74.6_{27.8}$ |
| OnlineContr | 0.50 | $89.7_{25.9}$ | $91.6_{24.4}$ | $88.2_{27.3}$ | $90.6_{26.0}$ | $86.0_{27.6}$ | $89.3_{27.2}$ | $79.0_{26.9}$ | $76.5_{27.9}$ |
| OnlineContr | 0.75 | $89.5_{26.5}$ | $91.7_{24.5}$ | $88.6_{27.4}$ | $90.8_{25.6}$ | $87.2_{27.9}$ | $89.2_{27.6}$ | $80.0_{27.4}$ | $77.5_{28.2}$ |
| OnlineContr | 1.00 | $89.6_{26.6}$ | $91.7_{25.0}$ | $88.3_{27.3}$ | $90.7_{26.0}$ | $87.5_{27.7}$ | $89.6_{27.5}$ | $80.5_{27.8}$ | $78.4_{28.7}$ |
| Triplet | 0.01 | $90.2_{25.6}$ | $91.5_{25.1}$ | $82.5_{25.7}$ | $90.3_{24.9}$ | $84.0_{25.5}$ | $89.1_{26.2}$ | $78.5_{24.5}$ | $76.9_{26.9}$ |
| Triplet | 0.10 | $\mathbf{90.6_{26.3}}$ | $\mathbf{91.9_{25.0}}$ | $\mathbf{89.7_{27.1}}$ | $\mathbf{91.2_{25.6}}$ | $\mathbf{88.4_{27.2}}$ | $\mathbf{89.9_{27.0}}$ | $83.5_{26.9}$ | $80.6_{28.6}$ |
| Triplet | 1.00 | $90.1_{25.7}$ | $90.9_{23.5}$ | $88.4_{26.6}$ | $90.6_{24.9}$ | $87.4_{27.0}$ | $88.6_{25.7}$ | $\mathbf{84.1_{28.6}}$ | $\mathbf{83.2_{31.1}}$ |
| Triplet | 5.00 | $88.2_{25.1}$ | $89.3_{23.4}$ | $86.5_{26.8}$ | $90.1_{25.1}$ | $84.9_{26.5}$ | $88.2_{26.1}$ | $81.5_{27.7}$ | $81.3_{30.1}$ |
| Triplet | 7.50 | $88.2_{25.4}$ | $89.6_{23.1}$ | $86.6_{27.0}$ | $90.1_{25.0}$ | $84.8_{26.4}$ | $88.2_{25.9}$ | $81.4_{27.8}$ | $81.5_{30.1}$ |
| Triplet | 10.00 | $88.1_{25.1}$ | $89.6_{22.9}$ | $86.8_{26.6}$ | $90.2_{24.9}$ | $84.8_{26.8}$ | $88.1_{26.2}$ | $81.6_{27.8}$ | $81.2_{30.4}$ |
| Average | - | **88.5** | **90.3** | 86.8 | 89.8 | 84.9 | 88.4 | 79.2 | 76.6 |

Table 6: Semantic similarity scores for all loss configurations after 5 epochs with $N = 50,000$ samples, retrieving $k = 16$ sentences.

| Loss | $\lambda$ | e5-small | | gte-base | | gte-small | | minilm-6 | |
|---|---|---|---|---|---|---|---|---|---|
| | | sarcastic | sst2 | sarcastic | sst2 | sarcastic | sst2 | sarcastic | sst2 |
| Reference | - | $83.4_{1.5}$ | $85.5_{1.7}$ | $81.4_{1.6}$ | $83.7_{1.4}$ | $82.5_{1.6}$ | $84.8_{1.4}$ | $42.3_{5.6}$ | $46.6_{7.4}$ |
| Cosine (SetFit) | - | $78.5_{2.1}$ | $81.6_{2.1}$ | $75.6_{3.0}$ | $79.9_{1.8}$ | $75.6_{2.5}$ | $80.7_{1.8}$ | $17.8_{5.6}$ | $27.1_{6.9}$ |
| Contrastive | 0.10 | $79.4_{2.0}$ | $83.3_{2.0}$ | $75.0_{2.4}$ | $81.0_{1.8}$ | $78.7_{2.1}$ | $82.1_{1.8}$ | $25.6_{6.6}$ | $34.8_{7.2}$ |
| Contrastive | 0.25 | $79.7_{2.0}$ | $83.7_{1.9}$ | $76.2_{2.4}$ | $81.4_{1.8}$ | $79.0_{2.1}$ | $82.6_{1.7}$ | $26.6_{6.6}$ | $34.5_{6.8}$ |
| Contrastive | 0.50 | $79.7_{2.0}$ | $83.8_{1.9}$ | $76.9_{2.4}$ | $81.6_{1.7}$ | $79.1_{2.1}$ | $82.8_{1.6}$ | $27.1_{6.6}$ | $34.2_{6.7}$ |
| Contrastive | 0.75 | $79.8_{2.0}$ | $83.8_{1.9}$ | $76.5_{2.6}$ | $81.5_{1.7}$ | $78.7_{2.3}$ | $82.7_{1.6}$ | $27.1_{6.5}$ | $34.1_{6.6}$ |
| Contrastive | 1.00 | $79.8_{2.0}$ | $83.7_{1.9}$ | $76.5_{2.7}$ | $81.3_{1.7}$ | $78.1_{2.5}$ | $82.4_{1.6}$ | $27.8_{6.5}$ | $33.9_{6.6}$ |
| MultipleNeg | - | $\mathbf{82.5_{1.6}}$ | $\mathbf{84.7_{1.8}}$ | $\mathbf{80.4_{1.8}}$ | $\mathbf{82.5_{1.6}}$ | $81.6_{1.8}$ | $\mathbf{83.9_{1.6}}$ | $\mathbf{39.9_{6.1}}$ | $\mathbf{43.5_{7.8}}$ |
| OnlineContr | 0.10 | $80.1_{1.9}$ | $83.8_{1.9}$ | $75.6_{2.3}$ | $81.2_{1.7}$ | $79.2_{2.0}$ | $82.5_{1.7}$ | $25.4_{6.7}$ | $33.2_{7.1}$ |
| OnlineContr | 0.25 | $80.5_{1.9}$ | $\mathbf{84.1_{1.9}}$ | $77.1_{2.3}$ | $81.7_{1.8}$ | $79.7_{1.9}$ | $82.9_{1.7}$ | $27.1_{6.6}$ | $33.0_{6.9}$ |
| OnlineContr | 0.50 | $80.6_{1.9}$ | $\mathbf{84.1_{1.9}}$ | $77.8_{2.3}$ | $82.0_{1.6}$ | $\mathbf{79.9_{2.0}}$ | $\mathbf{83.0_{1.6}}$ | $28.3_{6.5}$ | $33.9_{7.0}$ |
| OnlineContr | 0.75 | $80.6_{1.9}$ | $84.0_{1.9}$ | $77.5_{2.5}$ | $81.9_{1.6}$ | $79.4_{2.2}$ | $82.9_{1.6}$ | $28.5_{6.5}$ | $34.5_{7.0}$ |
| OnlineContr | 1.00 | $80.6_{1.9}$ | $84.0_{1.9}$ | $77.4_{2.6}$ | $81.7_{1.6}$ | $78.9_{2.3}$ | $82.7_{1.6}$ | $29.2_{6.4}$ | $35.0_{7.0}$ |
| Triplet | 0.01 | $81.2_{1.8}$ | $83.8_{2.0}$ | $78.0_{2.4}$ | $81.9_{1.7}$ | $\mathbf{79.9_{2.0}}$ | $\mathbf{83.0_{1.7}}$ | $25.8_{6.2}$ | $33.9_{7.3}$ |
| Triplet | 0.10 | $\mathbf{81.3_{1.7}}$ | $83.7_{1.9}$ | $\mathbf{78.1_{2.3}}$ | $81.9_{1.7}$ | $\mathbf{79.9_{2.1}}$ | $\mathbf{83.0_{1.6}}$ | $30.5_{6.1}$ | $35.2_{7.3}$ |
| Triplet | 1.00 | $79.2_{2.1}$ | $82.8_{2.1}$ | $76.3_{2.9}$ | $80.3_{1.8}$ | $77.2_{2.6}$ | $81.3_{1.6}$ | $23.7_{6.0}$ | $30.5_{7.0}$ |
| Triplet | 5.00 | $78.3_{2.1}$ | $81.8_{2.1}$ | $74.6_{2.7}$ | $79.9_{1.8}$ | $75.8_{2.7}$ | $80.6_{1.7}$ | $20.6_{5.9}$ | $29.6_{7.0}$ |
| Triplet | 7.50 | $78.4_{2.1}$ | $81.8_{2.1}$ | $74.7_{2.7}$ | $79.9_{1.8}$ | $75.7_{2.7}$ | $80.6_{1.7}$ | $20.4_{5.9}$ | $29.5_{7.0}$ |
| Triplet | 10.00 | $78.3_{2.1}$ | $81.8_{2.1}$ | $74.7_{2.7}$ | $80.0_{1.8}$ | $75.7_{2.7}$ | $80.7_{1.7}$ | $20.5_{5.9}$ | $29.6_{7.0}$ |
| Average | - | **80.0** | **83.5** | 76.7 | 81.3 | 78.6 | 82.3 | 26.7 | 33.7 |

MultipleNegativesRankingLoss may be unsuitable for fine-tuning toward other objectives as we have less control over the targeted separations between specific sentences. The other loss functions have separate example generation implementations with control over $\lambda$ parameter that defines the margin between similar and dissimilar sentences. Interestingly, independent of the loss function, this value does not necessarily correlate with good model performance. For distinguishing polarity, higher $\lambda$ values resulted in only slightly improved scores for the ContrastiveLoss. For TripletLoss, the opposite is true, contradicting the intuition that the margin of two embeddings in vector space should be separated *more* rather than less. The subtle differences between the embeddings may thus be small enough for larger margins to be impossible for specific configurations. As for the models, the *e5-small* scores highest for nearly all configurations, being effective at maximizing both polarity and semantic similarity, as is evident from the *average* row. For further details on model performance as of the

final experiment with $50,000$ samples, see Appendix B for the average score across all loss functions per model and Appendix C for details on each loss function, separated on both models and datasets. A final evaluation on the well established SentEval toolkit (Conneau & Kiela, 2018) allows us to compare our models on a series of tasks for the two best-performing baseline models (gte-base and e5-small). Table 7 shows the results of TripletLoss with a margin of 0.1 against a similar training procedure with the SetFit model, both trained with $50,000$ samples and sorted by average score. We reuse the suitable metrics from Reimers & Gurevych (2019) for fine-tuning on NLI data. Note how the fine-tuning approach achieves better overall scores and especially so for the MR (Movie Reviews) and SST-2. Our models also transfer well to tasks like SUBJ (subjective/objective classification). Comparing models of different loss functions is challenging due to the different data formats, as we cannot guarantee a direct comparison when the inputs are unequal. Unlike typical research on loss functions, we did not consider the loss values obtained during training or evaluation, as we find these uninformative in this context, i.e., balancing two possibly opposing objectives. However, we argue that our suggested metrics in Section 3.1 are reasonable and intuitive, and can likely be used for further studies on sentence embeddings.

Table 7: Performance for the best configuration and SetFit with SentEval.

|  | Model | Dataset | MR | CR | SUBJ | MPQA | SST2 | TREC | avg |
|---|---|---|---|---|---|---|---|---|---|
| Triplet$\lambda$0.1 | gte-base | sst2 | **89.31** | **89.27** | **92.91** | 85.95 | **93.19** | 80.80 | **85.50** |
| Triplet$\lambda$0.1 | gte-base | sarcastic | 84.33 | 88.82 | 92.82 | **88.04** | 90.83 | 88.40 | 85.01 |
| Triplet$\lambda$0.1 | e5-small | sst2 | 88.95 | 88.98 | 91.06 | 86.28 | 93.41 | 79.80 | 84.97 |
| SetFit | gte-base | sst2 | 84.30 | 88.85 | 90.91 | 86.08 | 89.18 | 86.00 | 84.27 |
| SetFit | e5-small | sst2 | 85.43 | 85.16 | 86.58 | 83.93 | 91.05 | 88.00 | 82.18 |
| SetFit | gte-base | sarcastic | 81.61 | 86.52 | 90.01 | 87.50 | 88.69 | 86.00 | 81.92 |
| SetFit | e5-small | sarcastic | 82.69 | 83.97 | 90.65 | 86.80 | 88.80 | **90.20** | 81.62 |
| Triplet$\lambda$0.1 | e5-small | sarcastic | 82.40 | 76.27 | 90.47 | 85.75 | 89.95 | 71.40 | 78.81 |

## 6 CONCLUSION AND FUTURE WORK

This paper has explored the potential of encoding polarity into sentence embeddings while retaining semantic similarity, done by fine-tuning models on data generated to suit the objectives of various sentence-transformers loss functions. We introduced two metrics to evaluate our results: the Polarity and Semantic Similarity Score. We conducted two main experiments. First, we investigated the importance of the number of sample sizes in our modeling scheme, finding that larger sample sizes from the generated data contribute positively towards both metrics. We used a suitable sample size in the second experiment and compared all model and loss function configurations. We found that 1) the *e5-small-v2* model outperformed the other baseline models tested (*gte-base*, *gte-small* and *all-MiniLM-L6-v2*), and 2) the TripletLoss, especially for lower $\lambda$ margins, had the overall best results. We conclude that fine-tuning the *e5-small* model with TripletLoss using the presented *example generation* with a margin parameter of $\lambda = 0.1$ is likely to yield an efficient and high-performing model for polarity-aware semantic retrieval – here evaluated on binary sentiment and sarcastic news headlines.

Future work consists of several paths for improvement: 1) With the suggested model configuration, a broader range of tasks can be experimented with the same fine-tuning approaches beyond sarcastic and sentiment-based data. 2) The example generation process can be extended to support multiclass inputs by one-vs-rest and other methods to manage multiple classes with a system designed for contrasting two samples. 3) Although our proposed metrics are a first step in assessing multiple objectives in this novel context, combining them better to represent the drift of the original semantic similarity remains an open question.

REPRODUCIBILITY STATEMENT

All code is available in an anonymous repository on the *Anonymous GitHub* page.[6] Results and corresponding tables and figures are programmatically generated for efficient reproduction. Sampling operations are fully deterministic, with the use of a defined random state. Source datasets are provided as used after initial preprocessing, and the experiments are logically structured in the source code. Some results are compiled from the resulting logs using *wandb* (both from the API and local run files), which cannot be included because of personal identifiers. However, code is provided to handle the resulting log files after training to ensure reproducibility. The necessary parsed and anonymized data to reproduce tables and figures is included.

ETHICS STATEMENT

We have reviewed the ICLR Code of Ethics, and can ensure that our work aligns with its guidelines. Datasets and the pre-trained sentence-transformer models utilized in our experiments are already public and readily available. The final system can be used for automatic retrieval, which may impose ethical concerns, especially when used for public-facing applications. One must thus consider privacy, bias, fairness, and potential misuse of the results.

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

## A  Training Details

### A.1  Configuration for Sample Size

Table 8: Configuration for the initial run of all models and parameters.

| Parameter | Value |
| --- | --- |
| Epochs | 5 |
| Batch Size | 64 |
| $k$ | 16 |
| Learning Rate | $3 \times 10^{-5}$ |
| Train Samples | {50, 500, 2000, 5000, 10000, 20000, 50000, 100000} |

### A.2  Compute and Training Time

The fine-tuning of a model scales linearly with $N$, averaging 15sec/epoch per 10,000 samples on an RTX4090. A model with $N = 100,000$ and 10 epochs is thus completed in 1500 seconds (25 minutes). This does not include time spent on evaluation per epoch, which requires embedding the test and training set with the fine-tuned models. The setup described by the seventeen loss configurations in Table 2 and the four baseline models in Table 1, amounts to 68 models to evaluate each epoch. For the initial experiment with the range of 8 different training samples, this results in 544 models (per dataset).

## B  Average Model Performance

Averaging the loss function scores per dataset for all models (Figures 4, 5, 6, 7), we can observe the general capability of each model, along with performance against its untrained baseline.

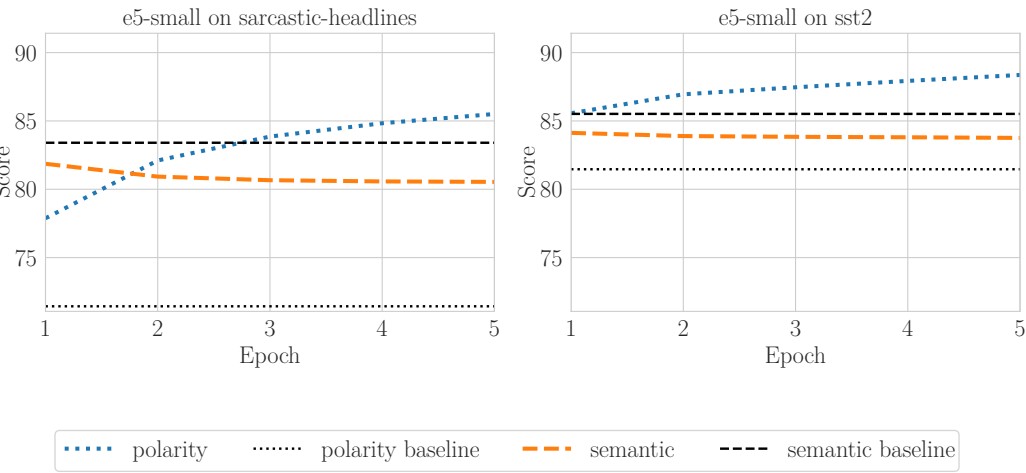

Figure 4: Average model performance for E5 Small

## C  Loss Performance per Model

Figures 8, 9, 10, 11 show the best-performing loss configuration per model. The strongest models all have acceptable performance for the selected loss functions. Triplet loss, however, stands out as it is only slightly below the baseline for the semantic similarity score and reaches the highest polarity scores. This is especially pronounced for the *e5-small* model.

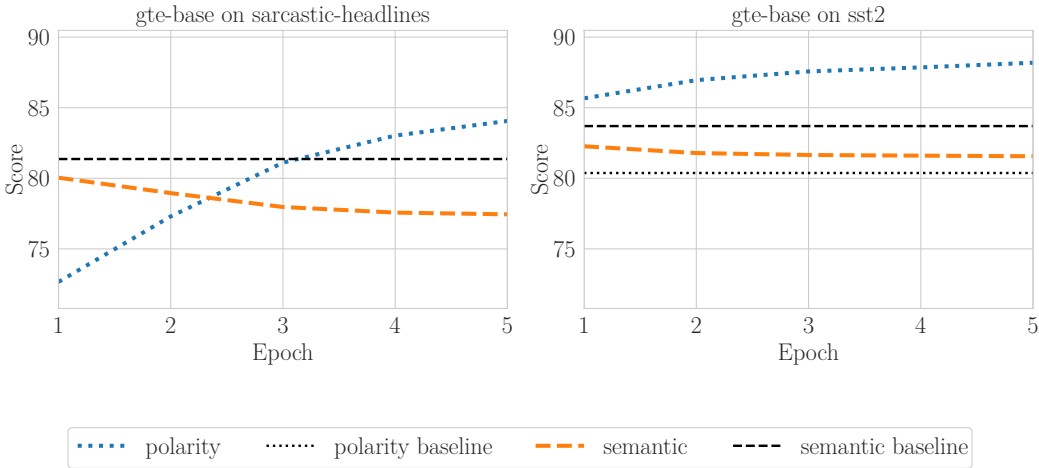

Figure 5: Average model performance for GTE base

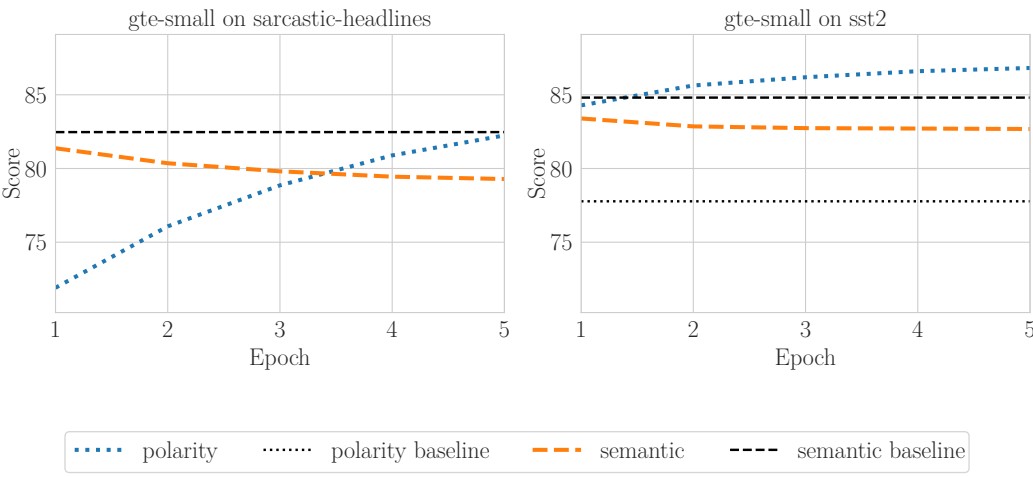

Figure 6: Average model performance for GTE small

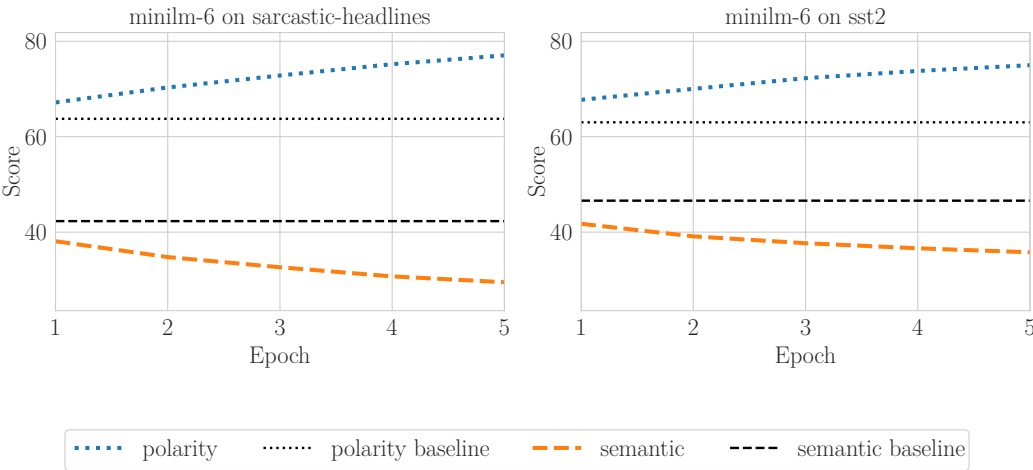

Figure 7: Average model performance for MiniLM-6

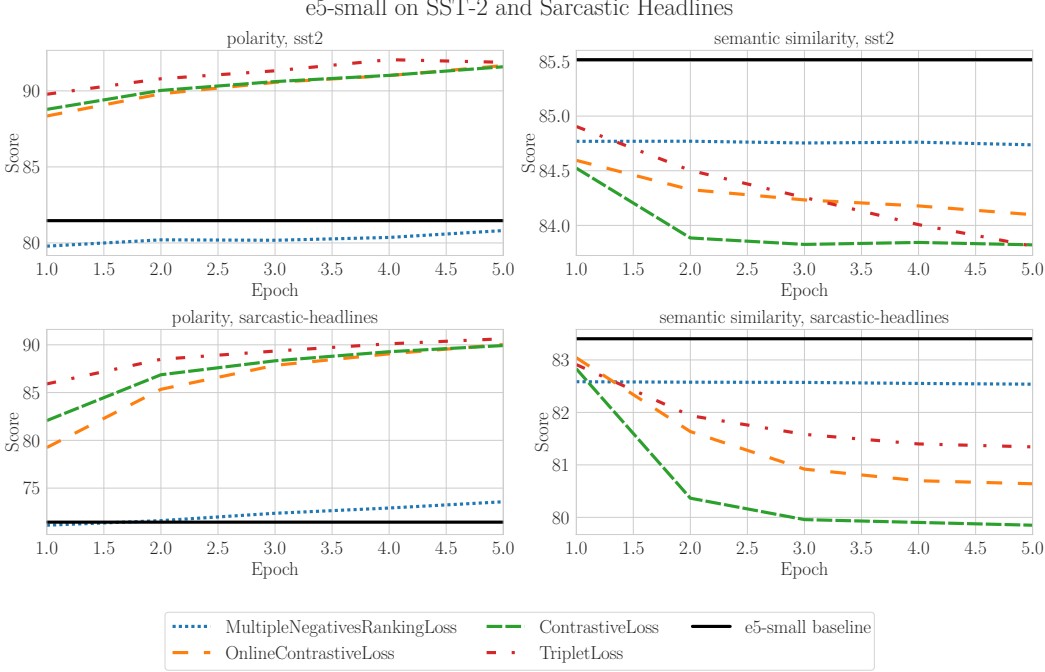

Figure 8: Max scores per loss for E5 Small

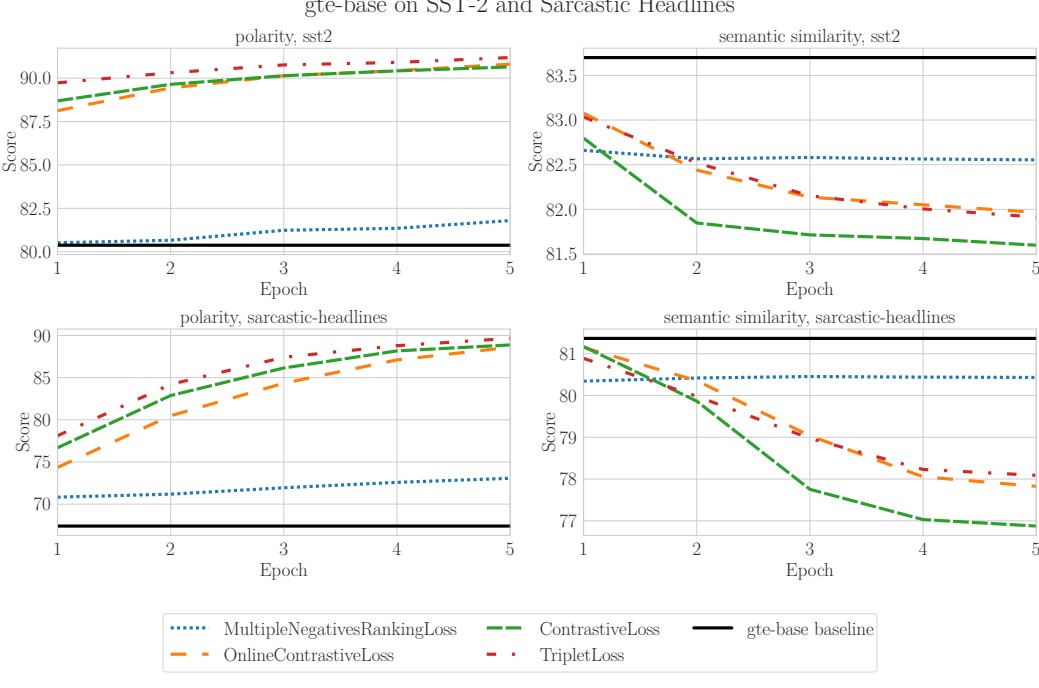

Figure 9: Max scores per loss for GTE base

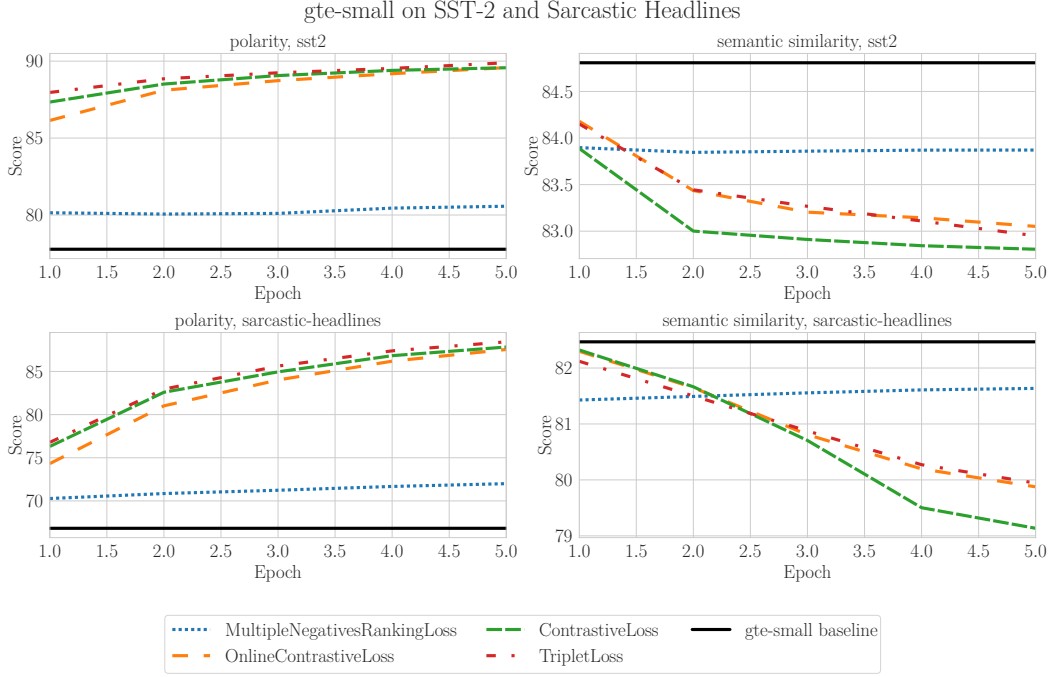

Figure 10: Max scores per loss for GTE small

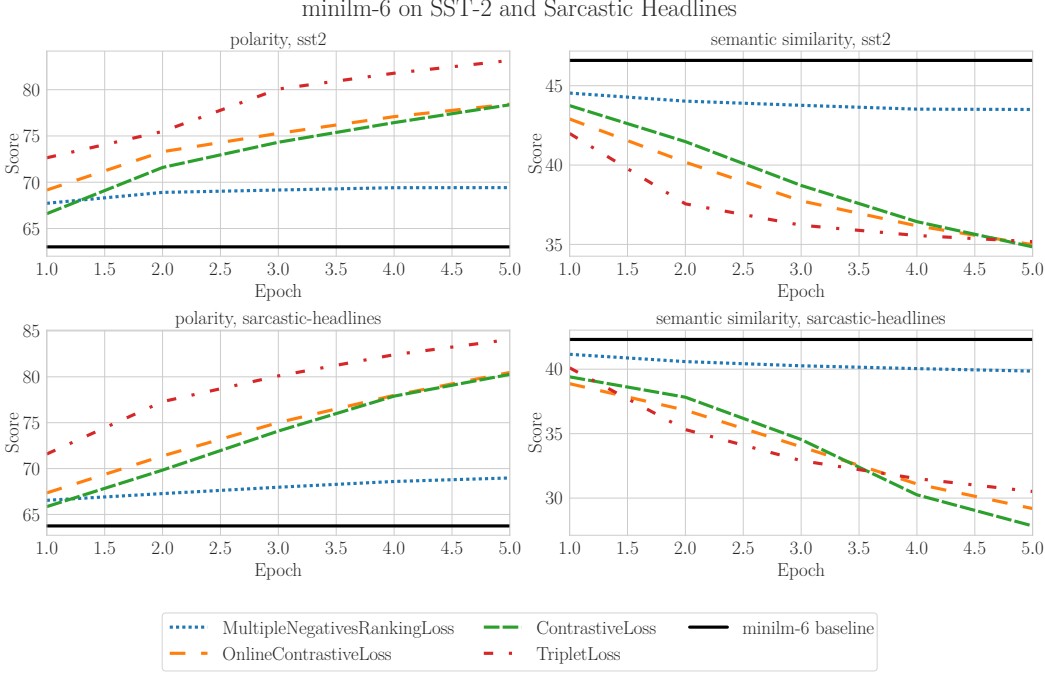

Figure 11: Max scores per loss for MiniLM-6

