# OpenReview forum: "Polarity-Aware Semantic Retrieval with Fine-Tuned Sentence Embeddings"
_ICLR.cc/2024/Conference — Submitted to ICLR 2024_

### Official Review · Reviewer_DEGg · 2023-10-28

**Soundness:** 1 poor
**Presentation:** 1 poor
**Contribution:** 1 poor
**Rating:** 3
**Confidence:** 3

**Summary:**

The paper studies the effectiveness of fine-tuning sentence embeddings for retrieving polarity-aware and semantic-aware similar sentences simultaneously. The authors propose metrics to measure polarity similarity and semantic similarity respectively. Experiments are done with four sentence embedding models of different sizes on two datasets.

**Strengths:**

1. The paper investigates the polarity capturing and semantic capturing of the finetuned sentence embedding methods.
2. The authors conduct experiments on two datasets with four base models.

**Weaknesses:**

1. The paper is poorly written, and the motivation and problem definition are not clear.
2. The problem studied is not insightful and the findings are trivial.
3. Many technique details are not well illustrated.

**Questions:**

1. Could you give a formal definition of the polarity? From my understanding, it is according to the label in the original dataset.
2. What is your motivation for this study and what is your takeaway conclusion? The experimental results are kind of trivial to me. “Polarity” and “Semantic” can have distribution overlap but are not exactly the same. So if you finetune the sentence embedding methods based on polarity perspective, it is quite common that it will suffer on the semantic dimension.
3. The definition of semantic score is confusing to me. Correct me if I am wrong, but it seems that if you finetune your sentence embedding models on whatever data, the semantic score will always decrease, since the finetuned model is diverging from the original parameters.
4. What is  w_i in sec 3.2.2? Is it the same as that in sec 3.2.1?
5. What is the reference model for semantic calculation used in sec 3.3 and sec 4? If I am understanding correctly, in section 4, you are using the base sentence embedding model without further finetuning as the reference model. Is it the same in sec 3.3?
6. How do you set \beta in sec 3.2.2?

---

> ### Author Response · Authors · 2023-11-20
>
> ## Weaknesses:
> - W1: I agree that our motivation and problem definition could be much better stated. And for that I thank you and some of the other reviewers in pointing out.
>
> - W2: There is a real use-case to retrieving with multiple objectives, and, as others have stated, relates to the catastrophic forgetting when fine-tuning these models. We showed that you can get a better trade-off than what is currently available with the architecture as-is, but by augmenting input data instead. I have since the reviews came in studied this in more depth and we out-perform alternatives such as SetFit in all related benchmarks with SentEval https://github.com/facebookresearch/SentEval, although we did not train for any of these objectives.
>
> - W3: This is valid, and we did not properly describe the weights (which are reused) and perhaps the failed attempt at averaging the metrics. We removed this and stuck to the two introduced metrics instead.
>
>
> ## Questions:
> - Q1: The polarity is the label. If this was unclear, we are strictly using binary labeled datasets.
>
> - Q2: The main motivation is rooted in a real-world application of e.g. retrieving K items that are similar on multiple objectives. An example would be a news article that has high similarity with another, while they may be connected to two entirely different topics (e.g. by sentiment). The entire point shown here is that we can increase the polarity far above the reduction of similarity, and this trade-off can be reduced by the configurations we evaluated.
>
> - Q3: Answered in Q2.
>
> - Q4: They are shared. We have since clarified this. The paper will be updated.
>
> - Q5: The reference model is always the respective baseline model (i.e. if the fine-tuned model is based on model A, the reference is A)
>
> - Q6: As mentioned, we opted for removing this common metric, as it provided little value in hindsight.

---

> > ### Comment · Reviewer_DEGg · 2023-11-22
> >
> > Thank you for your response. I have read the comments and decided to keep my score.

---

### Official Review · Reviewer_TX7z · 2023-10-30

**Soundness:** 1 poor
**Presentation:** 2 fair
**Contribution:** 1 poor
**Rating:** 3
**Confidence:** 5

**Summary:**

This paper aims to investigate the effect of fine-tuning sentence embeddings on two sentence retrieval tasks, equal polarity and high semantic similarity. This paper proposes two evaluation metric and conducts experiments on two sources of datasets.

**Strengths:**

1. This paper is generally self-contained for readers to understand different concepts. It provides enough tables and figures to illustrate text descriptions.

2. Experiment section contains enough implementation details, which allow readers to potentially reproduce the results in the paper.

3. This paper also identifies a few possible future works, which look promising and interesting.

**Weaknesses:**

Despite the above strengths, I still have the following concerns:

1. This paper lacks technical contribution. I can't see an originally proposed model architecture for sentence embedding problem. Instead, this paper looks more like a summary of why fine-tuning is an important technique for sentence embedding, but fine-tuning is also not a new concept and has been adopted by many previous works, including most LLMs.

2. When we do experiments, we usually need to repeat the same experiment multiple times and report both mean and standard deviation, in order to verify that the proposed model indeed significantly outperforms baselines. However, I can see mean but not standard deviation in the experiments.

**Questions:**

N.A.

---

> ### Author Response · Authors · 2023-11-20
>
> 1. I have questions regarding your definition of "technical contribution". While we do not propose a change in model architecture, we have shown that a change in the data, namely by augmenting input data on supervised data not related to sentence similarity, leads to improved retrieval scores for specific configurations. Fine-tuning on multiple objectives is vastly different from studying a single metric on existing datasets when fine-tuning. Otherwise, refer to the other comments.
>
> 2. Thanks for this. Will update the paper. From initial inspection, there are no changes to our conclusions based on the std, but is surely a much-needed addition.

---

### Official Review · Reviewer_Co32 · 2023-10-31

**Soundness:** 4 excellent
**Presentation:** 4 excellent
**Contribution:** 2 fair
**Rating:** 5
**Confidence:** 3

**Summary:**

This paper explores the efficacy of encoding polarity into sentence embeddings while
retaining semantic similarity, done by fine-tuning models on data generated to suit the objectives of various sentence-transformers loss functions. Two metrics were introduced to evaluate the results on The Stanford Sentiment Treebank binary classification and The News Headlines Dataset Sarcasm Detection: the Polarity and Semantic Similarity Score.

**Strengths:**

- Sound and rigorous experimentation, with reproducible results hosted from an anonymous repository
- Polarity scores for all loss configurations and Semantic similarity scores for all loss configurations and Aggregated scores across all configurations are reported meticulously
- Crystal clear presentation on the system, training configuration, data generation and tasks

**Weaknesses:**

The paper offers a crystal clear presentation of a system effectively fine-tuning sentence embeddings
for simultaneously retrieving sentences of equal polarity and high semantic similarity. However, in my opinion, it is not clear what the conclusion from the paper is, and whether anything *generalizable* can be drawn from the paper. For example, why SST-2 and Sarcastic Headlines are picked as the tasks for evaluation, why were the three loss functions picked, and most importantly, are not answered. In addition, aside from "To our knowledge, no comparable work exists to let us evaluate the model performance on this joint task with well-established metrics", no discussion on the extensibility of the work was offered. Perhaps the paper is better suited for a system track in an NLP conference, or a workshop in a machine learning conference, for its rigorous description of an interesting system offering 2 well-defined metrics for evaluation, and though its relative lack of contribution to machine learning.

**Questions:**

> We combine the two in a metric C, weighted with β (0.5 by default to produce the average),
primarily used to illustrate overall performance: Cβ(s) := β · PM(s) + (1 − β) · SM_M(s)

Why can the two metrics be combined in the form of a weighted sum? (After all, one may argue that the "units" of the two metrics are different.)

---

> ### Author Response · Authors · 2023-11-20
>
> Thanks for the detailed feedback, and sorry for the slow response.
>
> 1. I agree that we did not clearly present our motivation and conclusions. We've since updated it some.
>
> 2. The motivation was rooted in being able to retrieve similar and sarcastic sentences, and we later added SST-2 as it is a heavily used dataset for binary classification. I agree, though, that it should be connected to something more generalizable, and we have since studied its effects on a lot more tasks for sentence embeddings than what was presented in the paper by the SentEval library (https://github.com/facebookresearch/SentEval). Interestingly, our fine-tuning approach outperforms SetFit (the only comparable existing tool today) on every task.
>
> 3. There was a reason behind the selection of loss functions. This has been updated to account for every loss function mentioned in the sentence-transformers library.
>
> 4. Answer to 2 mentioned evaluation across different evaluations (subjectivity, movie reviews, opinion polarity, question types, etc.), which would likely build upon the extensibility of the work.
>
> 5. After reading the responses here, you are likely right on this one! Thank you regardless for the great insights provided by the review

---

> > ### Comment · Reviewer_Co32 · 2023-11-21
> > **Agreeing with another review on the lack of technical contribution**
> >
> > Clear and self-contained description of experiments, while the motivation for and conclusion from these experiments are murky.

---

### Official Review · Reviewer_urPj · 2023-11-01

**Soundness:** 3 good
**Presentation:** 2 fair
**Contribution:** 2 fair
**Rating:** 5
**Confidence:** 3

**Summary:**

This paper aimed to build models that retrieved similar sentences in both meaning and polarity. This paper addressed the problem of fine-tuning pre-trained sentence embedding models to a classification task while maintaining the ability to retrieve similar sentences. The authors presented two new metrics, the Polarity Score and the Semantic Similarity Score, to analyze the effect of fine-tuning using different contrastive-based losses. The authors argued that we can balance both STS (Semantic Similarity Score) and classification (Polarity Score) performances. To achieve this, the authors also proposed a new method to generate examples for fine-tuning. The example generations selected samples with similar meanings but opposite classification labels.

In the experiment, the author first established that the models achieved good scores for both semantic similarity and polarity without fine-tuning. The main results of the experiments showed that as we increased the number of samples, the model attained higher polarity scores, but the semantic similarity scores were slightly suffered. Upon further analysis, the authors suggested that the triplet loss performed best in achieving both scores and the margins (\lambda) did not have a large effect on performance.

**Strengths:**

1. This paper proposed a new formation of the text retrieval task.
2. This paper provided extensive experiment results on new retrieval metrics. The experiments included two datasets, and the results were consistent. It also confirmed previous findings that sentence-embedding models performed well in zero-shot classification and unsupervised retrieval.

**Weaknesses:**

Overall, I found the paper interesting, but some concerns regarding the significance of the proposed task and the results.

1. Although the paper proposed a new formulation of the text retrieval task, it did not mention why the task was important in the downstream applications or in further investigation of model capacity. Thus, the results of this paper provided limited implications.
2. In the direction of the model investigation, the experiment setup was based on a single dataset, i.e., fine-tuning and retrieving the same dataset. Consequently, the author could only conclude that the proposed method retained sentence similarity performance within the fine-tuned dataset. It is unclear how this is better than the standard benchmarks that included zero-shot classifications and unsupervised sentence retrieval.
3. Since the authors introduced the example generation approach, the results would be more meaningful if the authors compared with a baseline (original example generation.)
4. The problem addressed in this paper is relevant to the "forgetting effect" and its solutions, such as Chen et al., 2020 or Luo et al., 2023. The author should discuss this in the related work.


- Chen, S., Hou, Y., Cui, Y., Che, W., Liu, T., & Yu, X. (2020). Recall and learn: Fine-tuning deep pretrained language models with less forgetting. arXiv preprint arXiv:2004.12651.
- Luo, Y., Yang, Z., Meng, F., Li, Y., Zhou, J., & Zhang, Y. (2023). An empirical study of catastrophic forgetting in large language models during continual fine-tuning. arXiv preprint arXiv:2308.08747.

**Questions:**

1. Why did you train only 3 epochs to generate Figure 4?
2. Figure 4 was quite insightful, did the "max" models remain consistent?
3. Could you clarify why you did not test the fine-tuned models on STS using a standard benchmark?

---

> ### Author Response · Authors · 2023-11-20
>
> Sorry for the late response here.
>
> ## Weaknesses
>
> - W1/2. After the reviews came in, we attempted to make our statements more clear by its transferability to other applications, and did so using the established evaluations found in SentEval. We found our approach to outperform in all cases compared to e.g. SetFit on the same data with the same sample size. This should likely have been done beforehand, but we did not consider severely out-of-domain data as we were strictly focused on in-domain retrieval from an existing data source in the real world.
>
> - W3. This is something that went past us, and we have updated with the values from CosineSimilarity from the default SetFit configuration. Additional experiments show that all our model setups outperform SetFit by ~10% on average.
>
> - W4: Thanks for these resources.
>
>
> ## Questions
>
> - Q1: I'm sure you mean 5. Similar work shows that fine-tuning of transformer-based embedding models rarely require more than 3 epochs. As few signs of overfitting occur, we upped it to 5. We observe a steady increase (but very minor) > 5 epochs, which resulted in 10 for the final experiments.
>
> - Q2: The max models is the ideal result, and is nearly achieved by the best configurations shown in the end of the article, in table 7/8. We opted for changing this figure to box plots of similarity and polarity, giving more insights into the deviation in metrics per sample size.
>
> - Q3: As our goal is not generalized STS, I agree that we should have performed tests on established benchmarks, if not only for the sake of comparison to approach similar scores as the baseline while increasing scores on retrieval-based tasks within the same domains. As explained in comments below, we ended up rerunning experiments with our models for the tasks provided by SentEval (https://github.com/facebookresearch/SentEval) on various problems. Results are consistently better than the pre-trained alternatives for similar tasks like opinionated texts and reviews when fine-tuned on SST-2, and also outperforms the alternative SetFit model.

---

> > ### Comment · Reviewer_urPj · 2023-11-23
> > **Thank you for your response**
> >
> > The new results made the experiment results stronger, still, the motivation and the conclusion could be made more meaningful.

---

### Public Comment · ~Juri_Opitz2 · 2023-11-11
**Very interesting direction of work!**

Dear authors,

since sentence embeddings and "similarity" typically lack much interpretability, disentangling things like similarity and polarity really are good steps in the right direction! Since you mentioned the problem of catastrophic forgetting when learning the polarity, maybe you find a solution idea in our paper, where we learn a polarity embedding as part of the overall embedding. We find that the performance drop in similarity rating can be largely prevented by controlling the fine-tuning with a consistency loss.

[1] *SBERT studies Meaning Representations: Decomposing Sentence Embeddings into Explainable Semantic Features* https://arxiv.org/abs/2206.07023

---

> ### Author Response · Authors · 2023-11-11
>
> Thank you! I will definitely look into this. The concept of "decomposing" the embeddings is great, and I haven't seen this done anywhere else. I definitely think I can take some inspiration from the work you have done to study embeddings in more detail.

---

### Author Response · Authors · 2023-11-22

Thanks for the reviews. We have taken all feedback into consideration and have done minor rewriting and updates to generalized evaluations. These results much better reflect the effect of our defined metrics and loss configurations.

---

### Meta-Review · Area_Chair_gnTu · 2023-12-06

**Metareview:**

This paper proposes a new method for evaluating text retrieval by introducing two metrics, Polarity Score and Semantic Similarity score, to analyze the effect of fine-tuning using different contrastive losses. To balance the two metrics, the authors proposed a data augmentation method with pairs of similar meanings but opposite classification labels.

The reviewers have raised various concerns about the proposed methods, in particular:
- Lack of obvious downstream application (urPj)
- Limited choice of eval datasets (SST and sarcastic headlines detection), therefore unclear how generalizable the conclusions are (urPj, Co32)
- Limited technical contribution (TX7z)

**Justification For Why Not Higher Score:**

None of the reviewers seem to be excited about the contributions of the paper.

**Justification For Why Not Lower Score:**

N/A

---

### Decision · Program_Chairs · 2024-01-16

Reject